**Subject Area:**
cellular biology/molecular biology

chromosome segregation, DNA decatenation, DNA replication, Topoisomerase II, ultrafine anaphase bridge

**Authors for correspondence:**
Simon Gemble
e-mail: simon.gemble@curie.fr
Mounira Amor-Guéret
e-mail: mounira.amor@curie.fr

†Present address: Institut Curie, PSL Research University, CNRS, UMR 144, Biology of Centrosomes and Genetic Instability Laboratory, 75005, Paris, France.

# Topoisomerase IIα prevents ultrafine anaphase bridges by two mechanisms

Simon Gemble[1,2,3,†], Géraldine Buhagiar-Labarchède[1,2,3], Rosine Onclercq-Delic[1,2,3], Gaëlle Fontaine[1,2,3], Sarah Lambert[1,2,3] and Mounira Amor-Guéret[1,2,3]

[1]Institut Curie, PSL Research University, UMR 3348, Centre de Recherche, Orsay, France
[2]CNRS UMR 3348, Centre Universitaire, Bât. 110. 91405, Orsay, France
[3]Université Paris Saclay, UMR 3348, Centre Universitaire d'Orsay, France

SG, 0000-0002-4351-7136; GB-L, 0000-0002-2282-6693; RO-D, 0000-0001-7376-1095; GF, 0000-0002-7113-1687; SL, 0000-0002-1403-3204; MA-G, 0000-0002-7713-167X

Topoisomerase IIα (Topo IIα), a well-conserved double-stranded DNA (dsDNA)-specific decatenase, processes dsDNA catenanes resulting from DNA replication during mitosis. Topo IIα defects lead to an accumulation of ultrafine anaphase bridges (UFBs), a type of chromosome non-disjunction. Topo IIα has been reported to resolve DNA anaphase threads, possibly accounting for the increase in UFB frequency upon Topo IIα inhibition. We hypothesized that the excess UFBs might also result, at least in part, from an impairment of the prevention of UFB formation by Topo IIα. We found that Topo IIα inhibition promotes UFB formation without affecting the global disappearance of UFBs during mitosis, but leads to an aberrant UFB resolution generating DNA damage within the next G1. Moreover, we demonstrated that Topo IIα inhibition promotes the formation of two types of UFBs depending on cell cycle phase. Topo IIα inhibition during S-phase compromises complete DNA replication, leading to the formation of UFB-containing unreplicated DNA, whereas Topo IIα inhibition during mitosis impedes DNA decatenation at metaphase–anaphase transition, leading to the formation of UFB-containing DNA catenanes. Thus, Topo IIα activity is essential to prevent UFB formation in a cell-cycle-dependent manner and to promote DNA damage-free resolution of UFBs.

## 1. Introduction

Genome stability requires accurate DNA replication during S-phase and correct chromosome segregation during mitosis. Errors impairing these two crucial steps are particularly prone to induce genetic instability [1,2]. DNA replication leads to the formation of intertwines between two DNA strands, referred to as DNA catenanes, the resolution of which requires the introduction of transitory breaks. Topoisomerases play a key role in DNA catenane processing. Topoisomerase IIα (Topo IIα) is a well-conserved double-stranded DNA (dsDNA)-specific decatenase enzyme [2–6]. Topo IIα activity leads to double-strand breakage followed by intramolecular strand passage and DNA re-ligation [2]. The decatenating activity of Topo IIα plays a major role in several aspects of chromosome dynamics, including DNA replication and chromosome segregation [7].

Topoisomerase activity ahead of the replication fork cannot resolve all dsDNA catenanes. Moreover, convergence of two replisomes leads to the steric hindrance of topoisomerase activity [2,8]. Consequently, some dsDNA catenanes are not resolved before the onset of mitosis. They form physical links between the sister chromatids and must therefore be processed by Topo IIα before chromosome segregation in anaphase [9]. Indeed, the disruption of Topo IIα activity leads to incomplete sister chromatid disjunction [10,11]. Sister chromatid anaphase bridges are of two types: chromatin anaphase bridges that can be stained

with conventional dyes, such as DAPI, and ultrafine anaphase bridges (UFBs) that cannot be stained with conventional dyes or antibodies against histones. Both chromatin and ultrafine anaphase bridges result from a defect in sister chromatid segregation. During mitosis, PICH (Plk1-interacting checkpoint helicase), an SNF2-family DNA translocase involved in chromosome segregation [11–15], is recruited on both chromatin bridges and UFBs [10–14]. UFBs were discovered in 2007 [12,14] and are present in all cell lines tested. They are thus considered to be physiological structures [14,16]. Most UFBs are of centromeric origin, but some UFBs induced by replication stress originate from common fragile sites and are associated with FANCD2/FANCI proteins, whereas some other UFBs originate from telomeres or ribosomal DNA repeats [14,15,17,18]. UFBs were reported to contain either unresolved DNA catenations or replication intermediates [12–14,16,19]. Importantly, the total UFB population can be revealed only by PICH staining [12]. In a previous study, we reported that the intracellular accumulation of dCTP, due to cytidine deaminase (CDA) deficiency, leads to an excess of UFB-containing unreplicated DNA, due to a decrease in the basal activity of poly(ADP-ribose) polymerase 1 (PARP-1), which promotes the premature entry of cells into mitosis, before the completion of DNA replication has been completed [16,19].

Topo IIα inhibition leads to a large increase in the frequency of centromeric UFBs [11,12,14,20–22]. In this study, we investigated the molecular origin of the increase in UFB frequency following Topo IIα inhibition. We showed that Topo IIα inhibition had no effect on global disappearance of UFBs during mitotic progression. However, we observed an aberrant UFB resolution leading to DNA damage within the next G1 as revealed by the increase in the frequency of 53BP1 foci. We also found that Topo IIα inhibition led to two types of UFBs, the type of UFB formed depending on the phase of the cell cycle. Topo IIα inhibition during S-phase impairs DNA replication, leading to the formation of UFB-containing unreplicated DNA during mitosis, whereas Topo IIα inhibition during mitosis prevents DNA decatenation, resulting in UFB-containing dsDNA catenanes. Thus, Topo IIα inhibition impairs both DNA replication during S-phase and DNA decatenation during mitosis, leading to the formation of two types of UFB with different molecular origins. Our results therefore demonstrate that Topo IIα activity is required to prevent the formation of UFBs through replication defects or a lack of resolution of DNA catenanes when cells enter mitosis.

# 2. Results and discussion

## 2.1. Topo IIα inhibition promotes ultrafine anaphase bridge formation before and during mitosis

Topo IIα inhibition leads to a large increase in UFB frequency, and thus it has been proposed that Topo IIα activity is required for UFB resolution, accounting for the increase in UFB frequency upon Topo IIα inhibition [11,12,14,20–22]. However, the increase in UFB frequency upon Topo IIα inhibition could also reflect, at least in part, an accumulation of newly formed UFB. We therefore first investigated whether Topo IIα inhibition compromised the resolution or the formation of UFBs.

Topo IIα inhibition in HeLa cells with ICRF-159 (1 or 10 μM for 8 h), a catalytic Topo IIα specific inhibitor [23,24], led to an increase in UFB frequency in anaphase cells in a dose-dependent manner, as expected (figure 1a–c). We investigated whether Topo IIα inhibition affected UFB formation or resolution, by treating cells with ICRF-159 from S-phase until the end of mitosis. HeLa cell cycle duration is well described [25]: HeLa cells take about 8–10 h between S-phase and mitosis. Thus, mitotic cells after 8 h of ICRF-159 treatment correspond to cells that were in early S-phase when we added ICRF-159 to the cell culture medium (figure 1a). We then quantified PICH-positive UFBs from metaphase (the first step in mitosis, during which the distance between sister chromatids is sufficiently large for the visualization of UFBs) to telophase. Using this approach, we were able to assess UFB formation (by determining the increase in UFB frequency over time) and the global UFB disappearance (visualized as a decrease in UFB frequency during mitosis) (figure 1d).

ICRF-159 treatment led to an increase in UFB frequency at metaphase (red arrow, figure 1d). Thus, cells entered mitosis with a higher frequency of UFBs when Topo IIα was inhibited. Interestingly, the frequency of UFBs was also much higher at the metaphase–anaphase transition (green arrow, figure 1d), reflecting the formation of new UFBs early in mitosis. UFB frequency then decreased over time until the end of mitosis, in a similar manner in both untreated and ICRF-159-treated cells. UFBs are therefore resolved even if Topo IIα is inhibited. These results indicate that either Topo IIα activity is dispensable for the resolution of pre-existing UFBs during mitosis, or UFB dissolution is aberrant in the absence of Topo IIα activity, possibly through DNA breakage. However, our results demonstrate that Topo IIα activity is strictly necessary to prevent the formation of new UFBs. Our observations also indicate that Topo IIα inhibition promotes UFB formation in two different ways: before the onset of mitosis, as revealed by the increase in UFB frequency at metaphase, and during mitosis, leading to an increase in UFB frequency at the metaphase–anaphase transition.

## 2.2. Topoisomerase IIα inhibition impairs complete DNA replication

We previously reported that delaying entry into mitosis allows the completion of DNA replication and prevents the formation of UFBs, strongly suggesting that, in unchallenging condition, these structures result from the accumulation of unreplicated DNA during mitosis [16,19]. Topo IIα inhibition leads to an increase in UFB frequency at metaphase (figure 1d). We therefore first investigated whether Topo IIα inhibition prevented the completion of DNA replication, leading to the formation of new UFB-containing unreplicated DNA on entry into mitosis.

We therefore determined whether centromere replication was impaired upon Topo IIα inhibition. Cells were left untreated or were treated with ICRF-159 for 8 h. We used CREST staining to quantify double-dotted (yellow arrows, figure 2a) and single-dotted (white arrow, figure 2a) foci in prometaphase corresponding to fully replicated and unreplicated centromeres, respectively (figure 2a), as previously described [16]. The frequency of unreplicated centromeres was significantly higher in cells treated with 1 or 10 μM ICRF-159 for 8 h than in control cells (figure 2a and b), demonstrating that Topo IIα inhibition impaired the replication of centromeric DNA. Interestingly, the frequency of unreplicated centromeres did not differ between cells treated with 1 and 10 μM ICRF-159, contrasting with the dose-dependent effect of ICRF-159 on UFB formation (figure 1c).

royalsocietypublishing.org/journal/rsob    Open Biol. 10: 190259

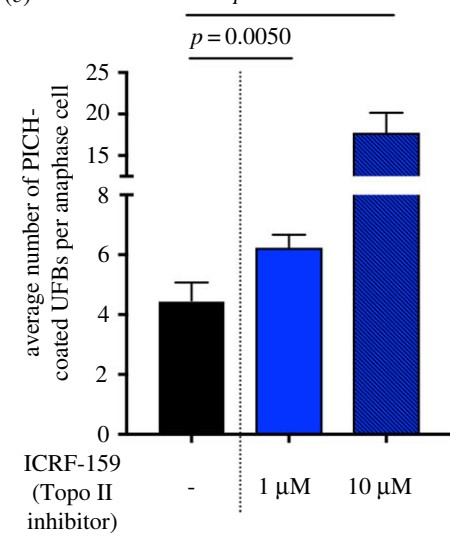

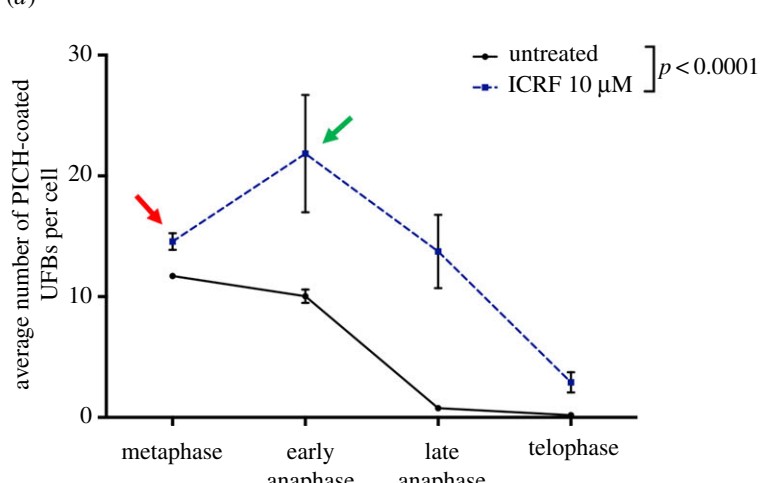

**Figure 1.** Topoisomerase IIα is not involved in UFB resolution. (*a*) Schematic representation of 8 h of Topo IIα inhibition during the cell cycle; only cells treated during S-phase to mitosis were analysed in anaphase. (*b*) Representative immunofluorescence deconvoluted *z*-projection images of PICH-positive UFBs in HeLa anaphase cells. DNA was visualized by DAPI staining (blue) and UFBs were stained with PICH antibody (in green). Enlarged image correspond to the yellow square. Scale bar: 5 µm. (*c*) Bar graph presenting the mean number of PICH-coated UFBs per anaphase cell in HeLa cells, for cells left untreated (black bar) or treated with 1 or 10 µM ICRF-159 for 8 h (blue bars); errors bars represent means ± s.d. from three independent experiments (50–100 anaphase cells analysed per condition). (*d*) Mean number of PICH-coated UFBs per mitotic cells, from metaphase to telophase, for cells left untreated (continuous line) or treated with 10 µM ICRF-159 for 8 h (discontinuous line); $n = 3$, more than 150 mitotic cells analysed per condition. Statistical significance was assessed in *t*-tests (*c*) or by two-way ANOVA (*d*).

For confirmation of the effect of Topo IIα inhibition on DNA replication, we then evaluated the levels of mitotic DNA synthesis (MiDAS). MiDAS contributes to the processing of unreplicated DNA sequences during mitosis and can therefore be used to detect problems leading to incomplete DNA replication during the previous S-phase [16,19,26–28]. MiDAS can be visualized by 5-ethynyl-2′-deoxyuridine (EdU) incorporation, leading to the formation of foci on condensed chromosomes (yellow arrow, figure 2*d*). We found that treatment with 1 or 10 µM ICRF-159 for 8 h led to a significant increase in the percentage of prometaphase cells presenting MiDAS, with no dose dependence (figure 2*c–e*). These data confirm that Topo IIα inhibition results in an accumulation of unreplicated DNA during mitosis, reflecting incomplete DNA replication in the previous S-phase. These observations are consistent with several studies in yeast or *in vitro*, showing that Topo IIα facilitates DNA replication [2,29–32]. They also suggest that Topo IIα activity is essential to promote complete DNA replication in mammalian cells.

Our data demonstrate that Topo IIα inhibition impairs the completion of DNA replication, probably leading to the formation of UFB-containing unreplicated DNA on entry into mitosis.

## 2.3. Topoisomerase IIα inhibition promotes the formation of two different types of ultrafine anaphase bridges depending on the phase of the cell cycle

Topo IIα inhibition increases total UFB frequency in a dose-dependent manner, but impairs DNA replication independently of ICRF-159 concentration, suggesting that Topo IIα inhibition affects another process of UFB formation, in addition to DNA replication. Topo IIα activity is required for both the completion of DNA replication during S-phase (figure 2) [2,29–32] and the DNA decatenation during mitosis [9]. We

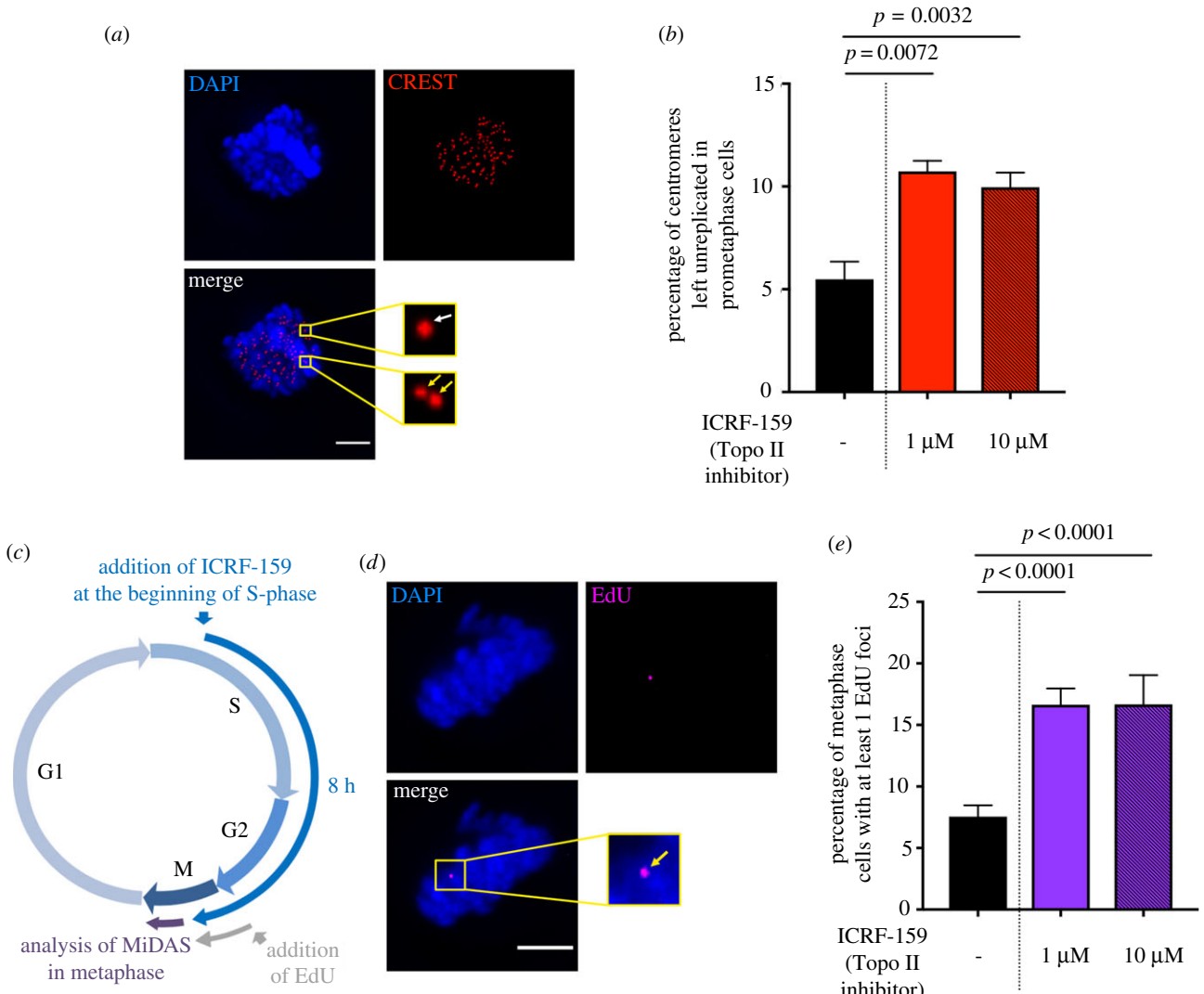

**Figure 2.** Topoisomerase IIα inhibition impairs complete DNA replication. (a) Representative immunofluorescence deconvoluted z-projection images of a prometaphase HeLa cell. DNA was visualized by DAPI staining (blue). Centromeres were stained with CREST serum (in red). Boxed images are enlarged; single-dotted CREST foci are indicated by white arrows and double-dotted CREST foci are indicated by yellow arrows. Scale bar: 5 µm. (b) Bar graph showing the percentage of centromeres left unreplicated in HeLa prometaphase cells left untreated (black bar) or treated with 1 or 10 µM ICRF-159 for 8 h (red bars). Error bars represent means ± s.d. from three independent experiments (more than 90 prometaphase cells per condition were analysed). (c) Schematic representation of 8 h of Topo IIα inhibition during the cell cycle. Only cells treated during S-phase to mitosis were analysed in anaphase. EdU was added 1 h before analysis. (d) Representative immunofluorescence deconvoluted z-projection images of a metaphase HeLa cell with EdU incorporation. DNA was visualized by DAPI staining (blue). EdU was stained with Alexa Fluor 555 (in magenta). Enlarged image shows one EdU focus on mitotic chromosomes (yellow arrow). Scale bar: 5 µm. (e) Bar graph presenting the percentage of HeLa metaphase cells presenting EdU foci after being left untreated (black bar) or after treatment with 1 or 10 µM ICRF-159 for 8 h (purple bars). Error bars represent means ± s.d. for three independent experiments (100–200 metaphase cells per condition were analysed). Statistical significance was assessed in t-test.

therefore investigated the respective contributions of these processes to the increase in UFB formation in response to Topo IIα inhibition. We analysed UFB frequency in HeLa anaphase cells after treatment either during S-phase (addition of ICRF-159 for 6 h followed by a release period of 3 h), or during mitosis (addition of ICRF-159 1 h before UFB analysis) (figure 3a). We confirmed the cell cycle phase specificity of our treatment by treating cells only during S-phase or only during mitosis (figure 3a), with EdU. As expected, all mitotic cells treated with ICRF-159 and EdU during S-phase were positive for EdU, whereas cells treated only during mitosis were EdU-negative (figure 3b). Consistent with these results, we found that Topo IIα inhibition during S-phase led to an increase in the percentage of unreplicated centromeres during mitosis and to an increase in the level of MiDAS (figure 3c and d), whereas inhibition during mitosis did not. These data confirm our previous findings (figure 2) and demonstrate the cell cycle

specificity of the treatment. These results indicate that Topo IIα activity is required during S-phase, to promote complete DNA replication.

In the same experimental conditions, Topo IIα inhibition during S-phase led to a slight, but significant, dose-independent increase in the mean number of UFBs per cell (figure 3b), probably due to the effect of Topo IIα inhibition on the completion of DNA replication (figures 2 and 3c,d). However, in cells treated with Topo IIα inhibitor only during mitosis, we observed a much higher dose-dependent increase in UFB frequency (figure 3b). These data suggest that most of the UFBs observed upon Topo IIα inhibition result from the loss of Topo IIα activity during mitosis.

We then investigated the effect of Topo IIα inhibition on the formation of different types of UFB as a function of the phase of the cell cycle. We treated cells with ICRF-159 (1 and 10 µM) during S-phase or during mitosis, and we analysed UFB

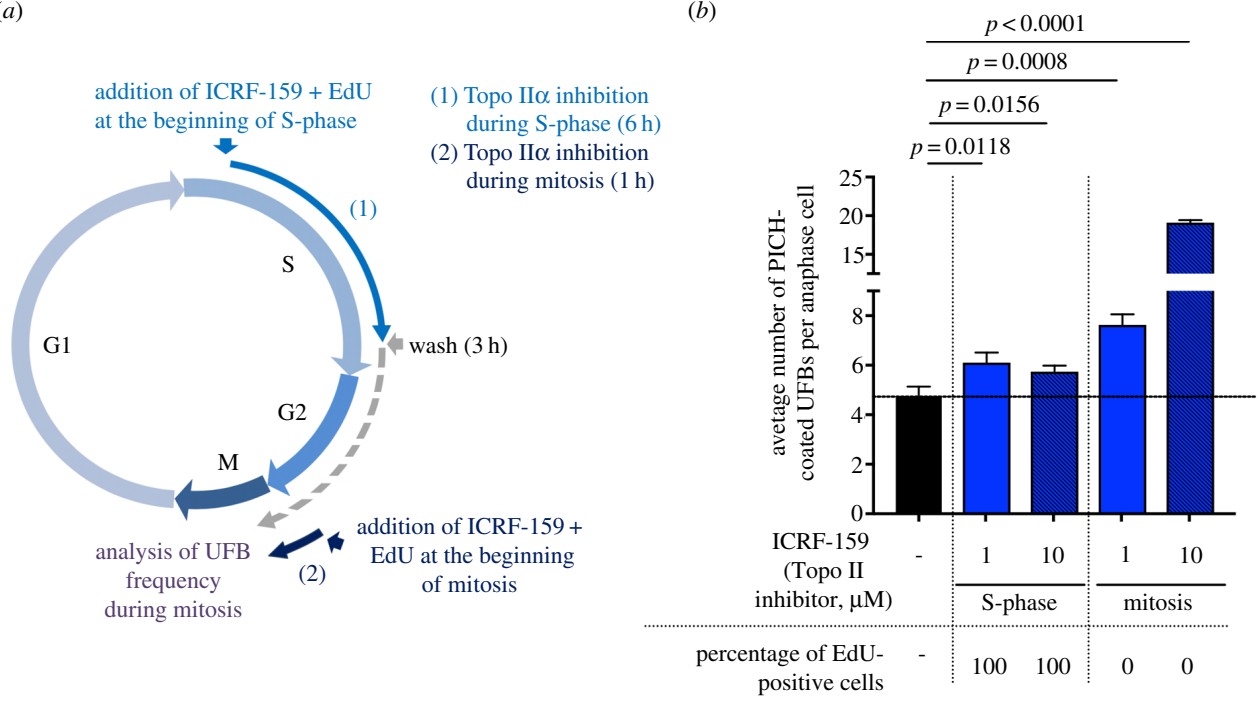

**Figure 3.** (*Caption overleaf.*)

royalsocietypublishing.org/journal/rsob   Open Biol. **10**: 190259

**Figure 3.** (*Overleaf*.) Topoisomerase IIα inhibition promotes two different types of UFB, depending of the phase of the cell cycle. (*a*) Schematic representation of Topo IIα inhibition during the cell cycle; only cells treated with ICRF-159 during S-phase (6 h) and then released (3 h) (i) or treated during mitosis (1 h) (ii) were analysed in anaphase. EdU was added with ICRF-159 to control cell cycle stage. (*b*) Bar graph showing the mean number of PICH-coated UFBs per anaphase cell in HeLa cells left untreated (black bar) or treated with 1 or 10 µM ICRF-159 during S-phase or during mitosis (blue bars). Percentages of EdU-positive cells for each condition are indicated below the graph. Errors bars represent means ± s.d. from three independent experiments (more than 85 anaphase cells analysed per condition). (*c*) Percentage of centromeres left unreplicated in HeLa prometaphase cells left untreated (black bar) or treated with 1 or 10 µM ICRF-159 during S-phase or mitosis (red bars). Error bars represent means ± s.d. from three independent experiments (more than 75 prometaphase cells per condition). (*d*) Percentage of HeLa metaphase cells presenting EdU foci after being left untreated (black bar) or after treatment with 1 or 10 µM ICRF-159 during S-phase or mitosis (purple bars). Error bars represent means ± s.d. for three independent experiments (more than 90 metaphase cells per condition were analysed). (*e*) Mean number of PICH-coated UFBs per mitotic cells, from metaphase to anaphase, for cells left untreated (continuous line) or treated with 1 or 10 µM ICRF-159 during S-phase (discontinuous lines; $n = 5$, 90–165 mitotic cells analysed per condition). (*f*) Mean number of PICH-coated UFBs per mitotic cells, from metaphase to anaphase, for cells left untreated (continuous line) or treated with 1 or 10 µM ICRF-159 during mitosis (discontinuous lines; $n = 5$, 90–165 mitotic cells analysed per condition). Statistical significance was assessed with *t*-test (*b*; *c* and *d*) or by two-way ANOVA test (*e* and *f*).

frequency in mitotic cells, from metaphase to anaphase (figure 3*e* and *f*). Topo IIα inhibition during S-phase led to an increase in the mean number of UFBs per cell at metaphase (red arrow, figure 3*e*), indicating that the cells entered mitosis with more UFBs. However, Topo IIα inhibition during S-phase was not associated with the formation of new UFBs at metaphase–anaphase transition (green arrow, figure 3*e*). UFB frequency decreased during the course of mitosis in both treated and untreated conditions. We therefore hypothesized that Topo IIα inhibition in S-phase would impair complete DNA replication, leading to the formation of UFB-containing unreplicated DNA on entry into mitosis. By contrast, the restriction of Topo IIα inhibition to mitosis had no effect on UFB frequency at metaphase (red arrow, figure 3*f*). However, UFB frequency was much higher at the metaphase–anaphase transition, particularly in response to 10 µM ICRF-159 (green arrow, figure 3*f*), reflecting the formation of new UFBs during mitosis. UFB frequency subsequently decreased during anaphase (figure 3*f*). Interestingly, the increase in UFB frequency at metaphase–anaphase transition was not observed in cells treated only during S-phase (green arrow, figure 3*e*). Topo IIα activity is required to resolve centromeric DNA catenations at the metaphase–anaphase transition [7]. We therefore suggest that DNA decatenation is compromised when Topo IIα is inhibited during mitosis, promoting the formation of UFB-containing DNA catenanes in anaphase. Consistent with this hypothesis, we observed that 66% of UFBs were of centromeric origin when Topo IIα was inhibited during mitosis (figure 3*b*; electronic supplementary material, figure S1A and B). This region of the chromosome has already been shown to be associated with UFB-containing DNA catenanes [11,14,21,22]. More importantly, UFB frequency decreased from early anaphase to late anaphase in cells treated with ICRF-159 during S-phase or during mitosis, despite the maintenance of Topo IIα inhibition during mitosis. These data indicate that UFBs are either resolved in absence of Topo IIα activity or probably broken due to aberrant resolution.

## 2.4. Topoisomerase IIα activity prevents DNA damage-associated resolution of ultrafine anaphase bridge during mitosis

To determine if Topo IIα activity is dispensable or not for UFB resolution during mitosis, we addressed the specific fate of UFBs. It has been reported that aberrant UFB resolution at the end of mitosis causes DNA damage leading to the formation of 53BP1 bodies in the next G1 phase, to protect

broken DNA ends until repair [33]. We investigated whether UFB resolution in cells treated with Topo IIα inhibitors was associated with DNA damage, by analysing the number of 53BP1 foci in the next G1 phase. Cells were synchronized by double thymidine block at the G1/S boundary and then released into cell cycle. Cells were left untreated or treated with ICRF-159 during S-phase or during mitosis (cell cycle distribution is shown in electronic supplementary material, figure S2A). First, to ensure that double thymidine block did not interfere with UFB formation when Topo IIα was inhibited, we analysed UFB frequency in synchronized cells treated with Topo IIα inhibitor during either S phase or mitosis, and we analysed the percentage of FANCD2-associated UFBs in these cells (electronic supplementary material, figure S2B and C). We confirmed an increase in UFB frequency in both cells treated during S-phase and during mitosis and found, as expected, that cells present a significant increase in the percentage of FANCD2-associated UFBs only when treated with Topo IIα inhibitor during S phase (electronic supplementary material, figure S2C). These results further support that the inhibition of Topo IIα during S-phase leads to unreplicated DNA marked by sister FANCD2 foci, but not when Topo IIα is inhibited in mitosis.

Then, we analysed 53BP1 foci in the next G1 phase of synchronized cells, left untreated or treated with 1 or 10 µM ICRF-159 during S-phase or during mitosis (figure 4*a–c*; electronic supplementary material, figure S2A). Topo IIα inhibition during S-phase led to a slight, but significant, dose-independent increase in the number of 53BP1 foci (figure 4*a–c*). However, in cells treated with Topo IIα inhibitor only during mitosis, we observed a much higher dose-dependent increase in 53BP1 foci. These results demonstrate that DNA damage in G1 was correlated to UFB frequency in the previous mitosis, meaning that more UFBs are formed in the previous mitosis, and more 53BP1 foci are generated in the next G1. These observations indicate aberrant UFB resolution during anaphase when Topo IIα is inhibited, probably by DNA breakage. These results confirm that Topo IIα activity is necessary for DNA damage-free resolution of UFB during mitosis.

Overall, our data shed light on the molecular origin of the supernumerary UFBs observed following Topo IIα inhibition, showing that they correspond to newly formed UFBs and probably not, or to a lesser extent, to unresolved pre-existing UFBs. Indeed, our data demonstrate that maintaining Topo IIα inhibition during mitosis does not affect the global UFB disappearance after metaphase–anaphase transition but lead to an accumulation of DNA damage in the next G1, reflecting aberrant UFB resolution. We concluded that Topo IIα activity

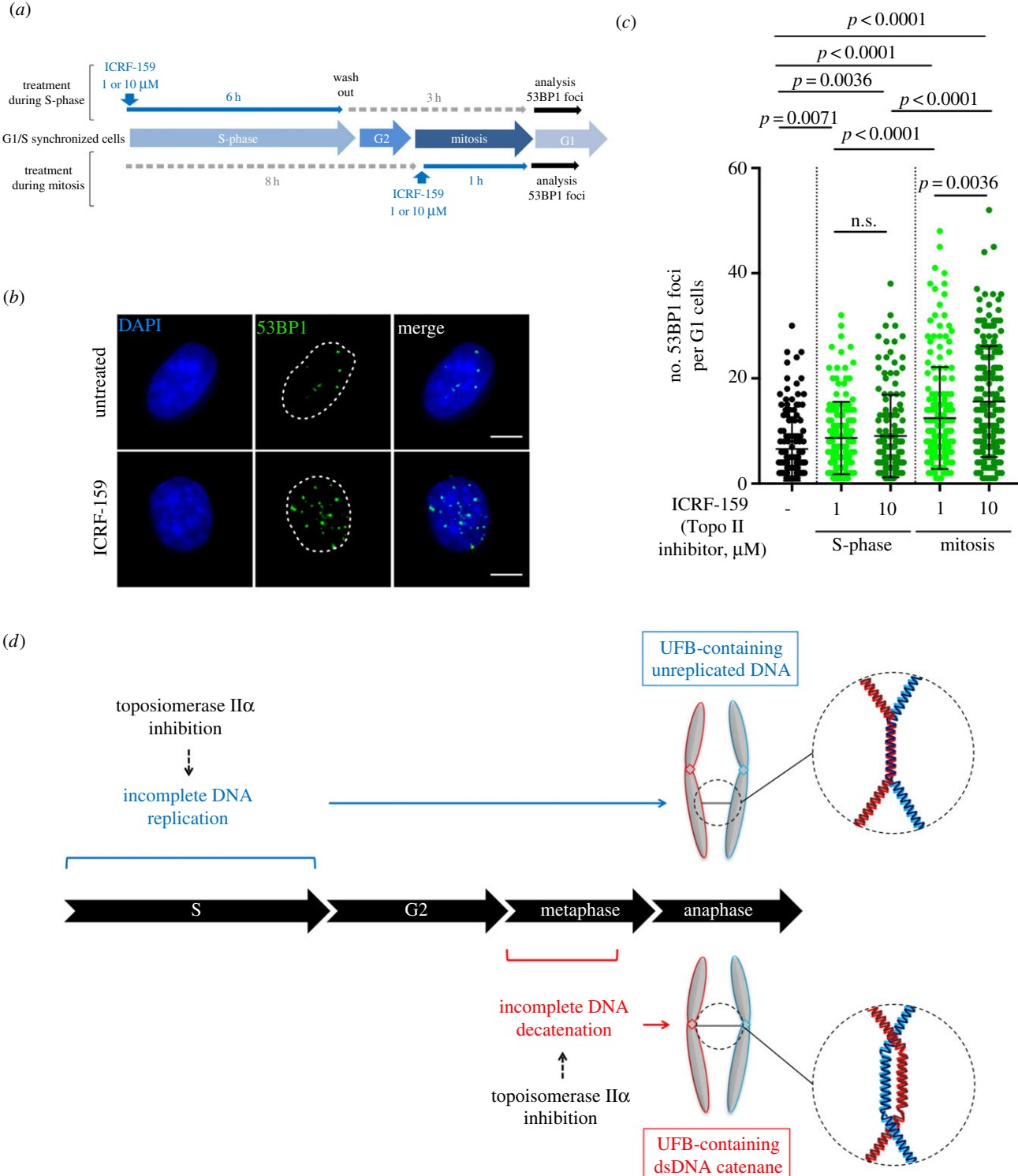

**Figure 4.** Topoisomerase IIα activity is necessary for UFB resolution. (*a*) Schematic representation of Topo IIα inhibition during the cell cycle. Cells were synchronized in G1/S boundary by a double thymidine block and then treated with ICRF-159 during S phase (6 h followed by 3 h washing) or during mitosis (1 h). (*b*) Representative immunofluorescence deconvoluted z-projection images of G1 HeLa cells. DNA was visualized by DAPI staining (blue). DNA damage was detected by staining with 53BP1 antibody (in green). Scale bar: 5 μm. (*c*) Dot plot presenting the number of 53BP1 foci per G1 HeLa cells, for cells left untreated (in black) or treated with 1 or 10 μM ICRF-159 during S-phase or during mitosis (in green); errors bars represent means ± s.d. from three independent experiments (more than 100 interphase cells analysed per condition). (*d*) Topo IIα inhibition leads to two types of UFBs, depending on the phase of the cell cycle. Topo IIα inhibition during S-phase compromises complete DNA replication, leading to the accumulation of unreplicated DNA in mitosis, resulting in an increase in the formation of UFB-containing unreplicated DNA. By contrast, Topo IIα inhibition during mitosis jeopardizes complete DNA decatenation process at the metaphase–anaphase transition, leading to the formation of UFB-containing DNA catenanes. Statistical significance was assessed in *t*-test.

in mitosis is necessary for the correct resolution of UFBs, as previously demonstrated [20].

We also found that Topo IIα inhibition during S-phase compromised the completion of DNA replication, leading to the accumulation of unreplicated DNA during mitosis, probably leading to the formation of UFB-containing unreplicated DNA. The restriction of Topo IIα inhibition to mitosis resulted in a much higher frequency of UFBs at the metaphase–anaphase transition, particularly in the presence of 10 μM ICRF-159, reflecting the formation of new UFBs during mitosis.

These UFBs probably result from impaired DNA decatenation at the metaphase–anaphase transition and correspond to newly formed UFB-containing DNA catenanes (figure 4d). Thus, our results indicate that the excess UFB observed upon Topo IIα inhibition results mainly from newly formed UFBs, in a replication- or decatenation-dependent manner, rather than from a delayed UFB resolution.

In conclusion, our findings show that Topo IIα is necessary to promote DNA damage-free resolution of UFBs and further extend the role of Topo IIα activity during the cell cycle, by showing that Topo IIα is required for complete DNA replication.

# 3. Material and methods

## 3.1. Cell culture and treatments

HeLa cells were cultured in DMEM supplemented with 10% FCS as previously described [16].

ICRF-159 (Razoxane) was provided by Sigma Aldrich (R8657) and was added to the cell culture medium at a final concentration of 1 or 10 μM following the protocol described in figures 1a, 2c, 3a, 4a and electronic supplementary material, figure S2A. Thymidine was provided by Sigma Aldrich (T9250) and was added to the cell culture medium at a final concentration of 2 mM.

All cells were routinely checked for mycoplasma infection.

## 3.2. Immunofluorescence microscopy

Immunofluorescence staining and analysis were performed as previously described [16]. Primary and secondary antibodies were used at the following dilutions: rabbit anti-PICH antibody (1 : 150; H00054821-D01P from Abnova); mouse anti-PICH antibody (1 : 400; H00054821-M01 from Abnova); human CREST antibody (1 : 100; 15-234-0001 from Antibodies Inc); rabbit anti-FANCD2 antibody (1 : 200; NB100-182 from Novus Biologicals); mouse anti-53BP1 antibody (1 : 500; MAB3802 from Millipore); Alexa Fluor 633-conjugated goat anti-human antibody (1 : 500; A21091 from Life Technologies); Alexa Fluor 555-conjugated goat anti-rabbit (1 : 500; A21429 from Life Technologies) and Alexa Fluor 555-conjugated goat anti-mouse (1 : 500; A21424 from Life Technologies). Cell images were acquired with a three-dimensional deconvolution imaging system consisting of a Leica DM RXA microscope equipped with a piezoelectric translator (PIFOC; PI) placed at the base of a 63x PlanApo N.A. 1.4 objective and a CoolSNAP HQ interline CCD camera (Photometrics). Stacks of conventional fluorescence images were collected automatically at a Z-distance of 0.2 μm (Metamorph software; Molecular Devices). Images are presented as maximum intensity projections, generated with ImageJ software, from stacks deconvolved with an extension of Metamorph software [34].

## 3.3. EdU staining

EdU incorporation into DNA was visualized with the Click-it EdU imaging kit (C10338 from Life Technologies), according to the manufacturer's instructions. EdU was used at a concentration of 2 μM for the indicated time. Cells were incubated with the click-it reaction cocktail for 15 min. Cell images were acquired with a three-dimensional deconvolution imaging system consisting of a Leica DM RXA microscope equipped with a piezoelectric translator (PIFOC; PI) placed at the base of a 63x PlanApo N.A. 1.4 objective and a CoolSNAP HQ interline CCD camera (Photometrics). Stacks of conventional fluorescence images were collected automatically at a Z-distance of 0.2 μm (Metamorph software; Molecular Devices). Images are presented as maximum-intensity projections generated with ImageJ software, from stacks deconvolved with an extension of Metamorph software.

## 3.4. Flow cytometry analysis

Cells were synchronized using double thymidine block: cells were incubated with 2 mM thymidine during 16 h and then released during 10 h in fresh medium and incubated again with 2 mM thymidine during 16 h. After ICRF-159 treatment, cells were detached by treatment with Accutase (Sigma), immediately washed in 1x PBS, fixed in 70% ethanol and stored at −20°C overnight. Cells were then washed twice with ice-cold 1x PBS and incubated with Vindelov solution (Tris HCl, pH 7.6 3,5 mM; NaCl 10 mM, propidium iodide 50 μg ml$^{-1}$; NP40 0.1%; RNAse 20 μg ml$^{-1}$) during 30 min in the dark. Finally, cell cycle analysis was analysed using FACSCanto II from BD Biosciences.

## 3.5. Statistical analysis

At least three independent experiments were carried out to generate each dataset and the statistical significance of differences was calculated with Student's t-test or two-way ANOVA, as indicated in figure legends.

Data accessibility. This article has no additional data.

Authors' contributions. S.G. performed the experiments, participated in the design of the experiments and data analysis, generated the figures and cowrote the manuscript. G.B.-L., G.F. and R.O.-D. performed experiments. S.L. contributed to data analysis and preparation of the manuscript. M.A.-G. supervised the study, analysed the data and cowrote the manuscript.

Competing interests. The authors declare that they have no conflict of interest.

Funding. This work was supported by grants from the Institut Curie (PICSysBio), the Centre National de la Recherche Scientifique (CNRS), the Ligue contre le Cancer (Comité de l'Essonne), the Association pour la Recherche sur le Cancer (ARC, SFI20121205645), the Institut National du Cancer (grant no. 2016-1-PLBIO-03-ICR-1) and by a fellowship awarded to S.G. by the Ministère de l'Education, de l'Enseignement Supérieur et de la Recherche and the ARC (DOC20140601310), and Institut Curie (PIC SysBio).

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
