## [Reviewer comments · Open Biology]

Review History

RSOB-19-0259.R0 (Original submission)

Review form: Reviewer 1

Recommendation

Major revision is needed (please make suggestions in comments)

Do you have any ethical concerns with this paper?

No

Comments to the Author

Topoisomerase IIA is a well-known enzyme to catalyse dsDNA (de)catenation. Its loss of function causes the formation and accumulation of DNA intertwining structures that compromise proper chromosome condensation and chromosome segregation in mitosis, where the latter manifests as chromatin and ultrafine DNA bridges (UFBs) in anaphase. The current study attempts to demonstrate that first, TOP2A is dispensable for UFB resolution in anaphase and secondly, its loss of function, depending on which cell cycle stage, can cause two different types of UFBs: unreplicated DNA bridges VS catenation bridges. I don't think there is enough and strong evidence to support the first claim (see below). Instead, I would suggest the authors focus on the

part of replication interference and unreplicated DNA bridge formation induced by ICRF159. Besides, the authors have to state clearly the details of UFBs analysis, e.g., were they counting centromere-specific UFBs or all UFBs in anaphase? Please also state what is metaphase DNA bridge, I assume they refer to those short PICH-coated threads between sister centromeres. The manuscript has to be substantially amended before publication.

Major concerns:

1. The authors claim that TOP2A inhibition, even throughout the entire mitosis, increases the number of UFBs but does not alter their resolution kinetics, suggesting that TOP2A is dispensable for their resolution. I don't think they can make such a strong claim. The disappearance of UFBs with anaphase-telophase progression could be a result of DNA breakage by mechanical forces or improperly cleaved by other nucleases. Or, though I think is unlikely, PICH may dissociate from UFBs during mitotic exit.

However, the authors try to demonstrate that without TOP2A in mitosis, there are no dose-dependent increases in DNA damage (due to chromatin bridges or UFBs) in G1 daughter cells induced by ICRF159 (Figure 4A-C). The main problem of this G1-damage assay is that their counting may be contaminated by other G1 cells that never experience TOP2A inhibition in mitosis. Figure 4A is misleading. Are they started with G1/S synchronised cells? I assume the authors use asynchronous cells treated with EdU for 1 hour, then wait for 8-10hours before the addition of ICRF159 for 1hour. Given that at the beginning there are already cells in mid, mid-late or late S-phase, these EdU-labelled populations will become G1 populations before the ICRF159 treatment. If this was the case, the authors were counting many untreated G1 cells. So the authors have to check and show what is the proportion of EdU+ve G1 cells before the addition of ICRF159? The results need to be subtracted by this population (they did not experience TOP2A inhibition). Besides, the authors did show there is an increase in DNA damage in the next G1 after TOP2A inhibition. This would suggest that TOP2A plays a role during chromosome segregation.

Therefore, although the claim is interesting, I don't think at the moment there is enough evidence to support it.

2. TOP2A inhibition in mitosis induces UFBs. In Figure 3F, it is not surprising to see overall increases in UFB formation in anaphase. However, it is misleading to compare between metaphase and anaphase because, in metaphase, the authors can only measure centromeric short UFBs whereas I assume the authors counted overall (CEN and arm) UFBs in anaphase. So I suggest the authors clearly state which UFBs they measure as that will change the interpretation significantly.

3. I think the whole study should be just focused on whether S-phase inhibition of TOP2A induced unreplicated DNA bridges in anaphase. The MiDAS data suggest that there is replication stress or underreplication. However, I am not convinced by the assay of unreplicated centromeres. Is this meant in untreated cells, there are always five centromeres unreplicated? Besides, I don't think the authors show MiDAS (EdU foci) at centromeres (Figure 2D & E). I think if the authors can show increased FANCD2 foci formation on mitotic cells and FANCD2-linked UFBs after S-phase inhibition of TOP2A that will further support the claim of arm underreplication. The part of centromere underreplication should be dropped out.

Therefore, I suggest the authors should focus on the findings of different UFBs formation after cell-cycle specific inhibition of TOP2A or TOP2A inhibition in S phase induces DNA underreplication and UFB formation. The claim that TOP2A is dispensable for UFB resolution should be dropped.

Minor concerns:

1. Can the authors try to explain why there is no dose-dependent increase in UFB formation when TOP2A is inhibited during S phase?
2. page 6, Gemble et al... ref format incorrect.
3. page 7, ...concentration, suggesting that
4. Figure 2E, y-axis percentage of metaphase cells have >1 EdU foci?
5. Figure 3A and 4A legends, please state clearly the treatment time and release time information.
6. Figure 3E, are they counting CEN-specific UFBs?

7. ICRF drugs can cause DNA damage by inhibiting TOP2B, so the authors should elaborate the possible reasons why ICRF159 treatment in S phase may induce incomplete DNA replication.

Review form: Reviewer 2

Recommendation

Major revision is needed (please make suggestions in comments)

Do you have any ethical concerns with this paper?

No

Comments to the Author

Manuscript ID: RSOB-19-0259

Title: TOPOISOMERASE II α PREVENTS, BUT DOES NOT RESOLVE, ULTRAFINE ANAPHASE BRIDGES BY TWO MECHANISMS

Authors: Gemble et al.,

It has been conventionally thought that since inhibition of Topoisomerase II α (Topo II α) leads to a large increase in Ultra Fine Bridges (UFBs), its activity was required for UFB resolution. However this manuscript by Amor-Gueret's group have put forward the hypothesis that the increase in UFB frequency upon TopoIIa inhibition, could be a result of just the accumulation of newly formed UFBs. The authors argue that lack of TopoIIa activity prevents UFB formation in a cell cycle specific manner and is not required for UFB resolution. The authors have argued out their case with a certain amount of conviction. The experiments are reasonably well done and controlled. The manuscript however requires the following modifications, additions before it becomes acceptable.

1. For all the figures the authors have used experiments based on cell cycle. Histograms should be provided for each experimental condition and treatment to show the cell cycle stage in which the experiments were conducted.
2. For the main datasets (i.e. the proof of principle experiments) – siRNA to TopoIIa should be conducted so that the data obtained with ICRF-159 can be validated.
3. The authors in Page 5 state: by treating the cells with ICRF-159 from S-phase until the end of mitosis". More clarity is required – which stage of S-phase? Early, middle or late? Exact number of hours of treatment should be indicated in the figure and in the histograms mentioned in point #1.
4. The authors have used two different concentrations of ICRF-159. Yet, except for UFB formation, two different concentrations of ICRF-159 have the same effect. The authors should either give a plausible reason or do the experiments using a gradient of ICRF-159 where differential effect will be visible.
5. For the amount of data presented in the manuscript, four Figures are not necessary. The authors should try to put their data in Figure 4 as part of Figure 3.
6. The authors should try to tighten the language in the text for Figure 3. It is extremely rambling and verbose for the moment.

Decision letter (RSOB-19-0259.R0)

09-Dec-2019

Dear Dr Gemble,

We are writing to inform you that the Editor has reached a decision on your manuscript RSOB-19-0259 entitled "TOPOISOMERASE II α PREVENTS, BUT DOES NOT RESOLVE, ULTRAFINE ANAPHASE BRIDGES BY TWO MECHANISMS", submitted to Open Biology.

As you will see from the reviewers' comments below, there are a number of criticisms that prevent us from accepting your manuscript at this stage. The reviewers suggest, however, that a revised version could be acceptable, if you are able to address their concerns. If you think that you can deal satisfactorily with the reviewer's suggestions, we would be pleased to consider a revised manuscript.

The revision will be re-reviewed, where possible, by the original referees. As such, please submit the revised version of your manuscript within four weeks. If you do not think you will be able to meet this date please let us know immediately.

When submitting your revised manuscript, please respond to the comments made by the referee(s) and upload a file "Response to Referees" in "Section 6 - File Upload". You can use this to document any changes you make to the original manuscript. In order to expedite the processing of the revised manuscript, please be as specific as possible in your response to the referee(s).

Please see our detailed instructions for revision requirements
<https://royalsociety.org/journals/authors/author-guidelines/>

Sincerely,
The Open Biology Team
<mailto:openbiology@royalsociety.org>

Associate Editor's Comments to Author(s):

Both referees recognised that the manuscript had potential and could possibly provide an interesting novel angle on topoisomerase II α function in mitosis. However, both referees also felt that the experimental descriptions were lacking critical details, particularly with regards to the cell cycle stage at which the cells for analysis were treated with ICRF159, and that the claim that topoisomerase II α is not required for UFB resolution in anaphase needed further experimental support. Both referees suggest a number of experiments to strengthen the manuscript, including further controls for the cell synchronisation and EdU labelling, confirmation of the effect of

TopoIIa inhibition by TopoIIa depletion and improved titration of ICRF159. The concerns of both referees should be addressed in a revised version of the manuscript.

Reviewer(s)' Comments to Author(s):

Referee: 1

Comments to the Author(s)

Topoisomerase IIA is a well-known enzyme to catalyse dsDNA (de)catenation. Its loss of function causes the formation and accumulation of DNA intertwining structures that compromise proper chromosome condensation and chromosome segregation in mitosis, where the latter manifests as chromatin and ultrafine DNA bridges (UFBs) in anaphase. The current study attempts to demonstrate that first, TOP2A is dispensable for UFB resolution in anaphase and secondly, its loss of function, depending on which cell cycle stage, can cause two different types of UFBs: unreplicated DNA bridges VS catenation bridges. I don't think there is enough and strong evidence to support the first claim (see below). Instead, I would suggest the authors focus on the part of replication interference and unreplicated DNA bridge formation induced by ICRF159. Besides, the authors have to state clearly the details of UFBs analysis, e.g., were they counting centromere-specific UFBs or all UFBs in anaphase? Please also state what is metaphase DNA bridge, I assume they refer to those short PICH-coated threads between sister centromeres. The manuscript has to be substantially amended before publication.

Major concerns:

1. The authors claim that TOP2A inhibition, even throughout the entire mitosis, increases the number of UFBs but does not alter their resolution kinetics, suggesting that TOP2A is dispensable for their resolution. I don't think they can make such a strong claim. The disappearance of UFBs with anaphase-telophase progression could be a result of DNA breakage by mechanical forces or improperly cleaved by other nucleases. Or, though I think is unlikely, PICH may dissociate from UFBs during mitotic exit.

However, the authors try to demonstrate that without TOP2A in mitosis, there are no dose-dependent increases in DNA damage (due to chromatin bridges or UFBs) in G1 daughter cells induced by ICRF159 (Figure 4A-C). The main problem of this G1-damage assay is that their counting may be contaminated by other G1 cells that never experience TOP2A inhibition in mitosis. Figure 4A is misleading. Are they started with G1/S synchronised cells? I assume the authors use asynchronous cells treated with EdU for 1 hour, then wait for 8-10hours before the addition of ICRF159 for 1hour. Given that at the beginning there are already cells in mid, mid-late or late S-phase, these EdU-labelled populations will become G1 populations before the ICRF159 treatment. If this was the case, the authors were counting many untreated G1 cells. So the authors have to check and show what is the proportion of EdU+ve G1 cells before the addition of ICRF159? The results need to be subtracted by this population (they did not experience TOP2A inhibition). Besides, the authors did show there is an increase in DNA damage in the next G1 after TOP2A inhibition. This would suggest that TOP2A plays a role during chromosome segregation.

Therefore, although the claim is interesting, I don't think at the moment there is enough evidence to support it.

2. TOP2A inhibition in mitosis induces UFBs. In Figure 3F, it is not surprising to see overall increases in UFB formation in anaphase. However, it is misleading to compare between metaphase and anaphase because, in metaphase, the authors can only measure centromeric short UFBs whereas I assume the authors counted overall (CEN and arm) UFBs in anaphase. So I suggest the authors clearly state which UFBs they measure as that will change the interpretation significantly.

3. I think the whole study should be just focused on whether S-phase inhibition of TOP2A induced unreplicated DNA bridges in anaphase. The MiDAS data suggest that there is replication stress or underreplication. However, I am not convinced by the assay of unreplicated centromeres. Is this meant in untreated cells, there are always five centromeres unreplicated?

Besides, I don't think the authors show MiDAS (EdU foci) at centromeres (Figure 2D & E). I think if the authors can show increased FANCD2 foci formation on mitotic cells and FANCD2-linked UFBs after S-phase inhibition of TOP2A that will further support the claim of arm underreplication. The part of centromere underreplication should be dropped out. Therefore, I suggest the authors should focus on the findings of different UFBs formation after cell-cycle specific inhibition of TOP2A or TOP2A inhibition in S phase induces DNA underreplication and UFB formation. The claim that TOP2A is dispensable for UFB resolution should be dropped.

Minor concerns:

1. Can the authors try to explain why there is no dose-dependent increase in UFB formation when TOP2A is inhibited during S phase?
2. page 6, Gemble et al... ref format incorrect.
3. page 7, ...concentration, suggesting that
4. Figure 2E, y-axis percentage of metaphase cells have >1 EdU foci?
5. Figure 3A and 4A legends, please state clearly the treatment time and release time information.
6. Figure 3E, are they counting CEN-specific UFBs?
7. ICRF drugs can cause DNA damage by inhibiting TOP2B, so the authors should elaborate the possible reasons why ICRF159 treatment in S phase may induce incomplete DNA replication.

Referee: 2

Comments to the Author(s)

Manuscript ID: RSOB-19-0259

Title: TOPOISOMERASE II α PREVENTS, BUT DOES NOT RESOLVE, ULTRAFINE ANAPHASE BRIDGES BY TWO MECHANISMS

Authors: Gemble et al.,

It has been conventionally thought that since inhibition of Topoisomerase II α (Topo II α) leads to a large increase in Ultra Fine Bridges (UFBs), its activity was required for UFB resolution. However this manuscript by Amor-Gueret's group have put forward the hypothesis that the increase in UFB frequency upon TopoIIa inhibition, could be a result of just the accumulation of newly formed UFBs. The authors argue that lack of TopoIIa activity prevents UFB formation in a cell cycle specific manner and is not required for UFB resolution. The authors have argued out their case with a certain amount of conviction. The experiments are reasonably well done and controlled. The manuscript however requires the following modifications, additions before it becomes acceptable.

1. For all the figures the authors have used experiments based on cell cycle. Histograms should be provided for each experimental condition and treatment to show the cell cycle stage in which the experiments were conducted.
2. For the main datasets (i.e. the proof of principle experiments) – siRNA to TopoIIa should be conducted so that the data obtained with ICRF-159 can be validated.
3. The authors in Page 5 state: by treating the cells with ICRF-159 from S-phase until the end of mitosis". More clarity is required – which stage of S-phase? Early, middle or late? Exact number of hours of treatment should be indicated in the figure and in the histograms mentioned in point #1.
4. The authors have used two different concentrations of ICRF-159. Yet, except for UFB formation, two different concentrations of ICRF-159 have the same effect. The authors should either give a

plausible reason or do the experiments using a gradient of ICRF-159 where differential effect will be visible.

5. For the amount of data presented in the manuscript, four Figures are not necessary. The authors should try to put their data in Figure 4 as part of Figure 3.

6. The authors should try to tighten the language in the text for Figure 3. It is extremely rambling and verbose for the moment.

Author's Response to Decision Letter for (RSOB-19-0259.R0)

See Appendix A.

RSOB-19-0259.R1 (Revision)

Review form: Reviewer 1

Recommendation

Accept with minor revision (please list in comments)

Do you have any ethical concerns with this paper?

No

Comments to the Author

The authors have carried out a new and important experiment to re-test if TOP2A is dispensable for proper UFB resolution during mitosis. In contrast to their previous conclusion, they now demonstrate that TOP2A is required for proper or damage-free UFB resolution, as published in other previous studies.

Regarding to another claim that TOP2A didn't affect the kinetics of UFB resolution, I found that their data are not consistently supporting it. In fig.1D, ICRF caused increases of UFB formation in early anaphase (~21/cell), which reduced to ~13/cell in late anaphase, indicating that cells cannot timely resolve all UFBs without TOP2A. However, in figs 3E and F, both showed a very sharp reduction of UFBs in late anaphase, arguing that TOP2A is dispensable. In fig. 3F, even cells have a higher number of UFBs in early anaphase (~29/cell), the drop is even more obvious (to ~6/cell in late anaphase). According to the result of fig. 1D, it suggests that there is a defect in UFB resolution kinetics whereas not from figs 3E and F.

On one hand, the authors now show TOP2A confers proper UFB resolution during mitosis but on the other hand, they claim the lack of TOP2A does not affect the kinetics of UFB resolution/disappearance (data seem not consistently support it). It is very hard to rationalize their conclusion. The authors should give a more reasonable explanation, or their interpretation may be incorrect.

Besides, the authors assume that the counting of metaphase UFBs can be used to represent the global UFB population in anaphase. It is totally misleading. As mentioned before, in metaphase where sister chromatids cohesion remains present, UFB can only be visualised at centromeres, so it is impossible to see catenane UFBs on arms. In anaphase where sister chromatid arms are separated, the authors can count all UFBs both from arms and centromeres. Therefore, it is inappropriate to compare UFB formation and resolution kinetics between metaphase and anaphase cells, unless they only count centromeric UFBs in anaphase; however, they didn't do it

in Fig. 1D, 3E, 3F. So, the kinetics can only be determined from early anaphase to telophase. Please provide evidence and image examples showing where UFBs form in metaphase cells, ideally with a CREST staining.

In page 9 (Fig. S1B), it is inaccurate to claim that most of UFBs induced by ICRF are of centromeric origin. The treatment of ICRF induces overall numbers of UFBs (both on arms and centromeres). The UFB CEN to ARM ratio only slightly increased and is not in a dose-dependent manner. Overall, I do not totally agree with the interpretation of some data and I think the authors should revise their claims carefully before accepted for publication.

Review form: Reviewer 2

Recommendation

Accept as is

Do you have any ethical concerns with this paper?

No

Comments to the Author

None

Decision letter (RSOB-19-0259.R1)

30-Mar-2020

Dear Dr Gemble

We are pleased to inform you that your manuscript RSOB-19-0259.R1 entitled "TOPOISOMERASE II α PREVENTS ULTRAFINE ANAPHASE BRIDGES BY TWO MECHANISMS" has been accepted by the Editor for publication in Open Biology. The reviewer(s) have recommended publication, but also suggest some minor revisions to your manuscript. Therefore, we invite you to respond to the reviewer(s)' comments and revise your manuscript.

Please submit the revised version of your manuscript within 7 days. If you do not think you will be able to meet this date please let us know immediately and we can extend this deadline for you.

- 1) A text file of the manuscript (doc, txt, rtf or tex), including the references, tables (including captions) and figure captions. Please remove any tracked changes from the text before submission. PDF files are not an accepted format for the "Main Document".
- 2) A separate electronic file of each figure (tiff, EPS or print-quality PDF preferred). The format should be produced directly from original creation package, or original software format. Please note that PowerPoint files are not accepted.
- 3) Electronic supplementary material: this should be contained in a separate file from the main text and meet our ESM criteria (see <http://royalsocietypublishing.org/instructions-authors#question5>). All supplementary materials accompanying an accepted article will be treated as in their final form. They will be published alongside the paper on the journal website and posted on the online figshare repository. Files on figshare will be made available approximately one week before the accompanying article so that the supplementary material can be attributed a unique DOI.

Online supplementary material will also carry the title and description provided during submission, so please ensure these are accurate and informative. Note that the Royal Society will not edit or typeset supplementary material and it will be hosted as provided. Please ensure that the supplementary material includes the paper details (authors, title, journal name, article DOI). Your article DOI will be 10.1098/rsob.2016[*last 4 digits of e.g. 10.1098/rsob.20160049*].

- 4) A media summary: a short non-technical summary (up to 100 words) of the key findings/importance of your manuscript. Please try to write in simple English, avoid jargon, explain the importance of the topic, outline the main implications and describe why this topic is newsworthy.

Images

Data-Sharing

It is a condition of publication that data supporting your paper are made available. Data should be made available either in the electronic supplementary material or through an appropriate repository. Details of how to access data should be included in your paper. Please see <http://royalsocietypublishing.org/site/authors/policy.xhtml#question6> for more details.

Data accessibility section

Sincerely,

The Open Biology Team
<mailto:openbiology@royalsociety.org>

Reviewer(s)' Comments to Author:

Referee: 2

Comments to the Author(s)
 None

Referee: 1

Comments to the Author(s)

The authors have carried out a new and important experiment to re-test if TOP2A is dispensable for proper UFB resolution during mitosis. In contrast to their previous conclusion, they now demonstrate that TOP2A is required for proper or damage-free UFB resolution, as published in other previous studies.

Regarding to another claim that TOP2A didn't affect the kinetics of UFB resolution, I found that their data are not consistently supporting it. In fig.1D, ICRF caused increases of UFB formation in early anaphase (~21/cell), which reduced to ~13/cell in late anaphase, indicating that cells cannot timely resolve all UFBs without TOP2A. However, in figs 3E and F, both showed a very sharp reduction of UFBs in late anaphase, arguing that TOP2A is dispensable. In fig. 3F, even cells have a higher number of UFBs in early anaphase (~29/cell), the drop is even more obvious (to ~6/cell in late anaphase). According to the result of fig. 1D, it suggests that there is a defect in UFB resolution kinetics whereas not from figs 3E and F.

On one hand, the authors now show TOP2A confers proper UFB resolution during mitosis but on the other hand, they claim the lack of TOP2A does not affect the kinetics of UFB resolution/disappearance (data seem not consistently support it). It is very hard to rationalize their conclusion. The authors should give a more reasonable explanation, or their interpretation may be incorrect.

Besides, the authors assume that the counting of metaphase UFBs can be used to represent the global UFB population in anaphase. It is totally misleading. As mentioned before, in metaphase where sister chromatids cohesion remains present, UFB can only be visualised at centromeres, so it is impossible to see catenane UFBs on arms. In anaphase where sister chromatid arms are separated, the authors can count all UFBs both from arms and centromeres. Therefore, it is inappropriate to compare UFB formation and resolution kinetics between metaphase and anaphase cells, unless they only count centromeric UFBs in anaphase; however, they didn't do it in Fig. 1D, 3E, 3F. So, the kinetics can only be determined from early anaphase to telophase. Please provide evidence and image examples showing where UFBs form in metaphase cells, ideally with a CREST staining.

In page 9 (Fig. S1B), it is inaccurate to claim that most of UFBs induced by ICRF are of centromeric origin. The treatment of ICRF induces overall numbers of UFBs (both on arms and centromeres). The UFB CEN to ARM ratio only slightly increased and is not in a dose-dependent manner. Overall, I do not totally agree with the interpretation of some data and I think the authors should revise their claims carefully before accepted for publication.

Author's Response to Decision Letter for (RSOB-19-0259.R1)

See Appendix B.

Decision letter (RSOB-19-0259.R2)

14-Apr-2020

Dear Dr Gemble

We are pleased to inform you that your manuscript entitled "TOPOISOMERASE II α PREVENTS ULTRAFINE ANAPHASE BRIDGES BY TWO MECHANISMS" has been accepted by the Editor for publication in Open Biology.

Article processing charge

Please note that the article processing charge is immediately payable. A separate email will be sent out shortly to confirm the charge due. The preferred payment method is by credit card; however, other payment options are available.

Sincerely,

The Open Biology Team

mailto: openbiology@royalsociety.org

Appendix A

Response for the revision of the article:

Manuscript ID: RSOB-19-0259

Title: TOPOISOMERASE II α PREVENTS ULTRAFINE ANAPHASE BRIDGES BY TWO
MECHANISMS

Authors: Gemble et al.,

Corresponding authors: mounira.amor@curie.fr and simon.gemble@curie.fr

Dear Editor,

We would like to thank you and to thank Open Biology editorial board for overseeing our manuscript.

After careful assessment of the reviewer's comments, we concluded that several points raised by the reviewers are due to a lack of sufficient information in the main text. As consequence, in this revised version of the manuscript we modified the text to clarify our data and discuss the comments raised by the two reviewers (in particular the comment related with the experimental approach to inhibit Topo II α during either S-phase or mitosis). Additionally, we performed new experiments to further demonstrate our model and to specifically answer to the reviewer's comments. In particular, as suggested by reviewer 1, we determined whether Topo II α activity was dispensable or not for UFB resolution during mitosis by analyzing the number of 53BP1 foci in cells synchronized by a double thymidine block at G1/S boundary, left untreated or treated with Topo II α inhibitor during S-phase or during mitosis (Figure 4 and S2A of the revised manuscript). Thanks to this experiment, we demonstrated an aberrant UFB resolution when Topo II α was inhibited during mitosis, confirming that Topo II α was necessary for UFB resolution during mitosis: the text was modified accordingly.

We also added an new author, Gaëlle Fontaine, who performed part of the experiments to answer reviewer's comments.

Below you will find our detailed response for the revision of the article. Our comments are in blue italic.

We hope that our strategy to satisfy the reviewer's comments will be accepted by Open Biology.

We are looking forward to hear from you soon,

With kind regards,

Mounira Amor-Gu ret and Simon Gemble

Referee 1:

Comments to the Author(s)

Topoisomerase IIA is a well-known enzyme to catalyse dsDNA (de)catenation. Its loss of function causes the formation and accumulation of DNA intertwining structures that compromise proper chromosome condensation and chromosome segregation in mitosis, where the latter manifests as chromatin and ultrafine DNA bridges (UFBs) in anaphase. The current study attempts to demonstrate that first, TOP2A is dispensable for UFB resolution in anaphase and secondly, its loss of function, depending on which cell cycle stage, can cause two different types of UFBs: unreplicated DNA bridges VS catenation bridges. I don't think there is enough and strong evidence to support the first claim (see below). Instead, I would suggest the authors focus on the part of replication interference and unreplicated DNA bridge formation induced by ICRF159. Besides, the authors have to state clearly the details of UFBs analysis, e.g., were they counting centromere-specific UFBs or all UFBs in anaphase? Please also state what is metaphase DNA bridge, I assume they refer to those short PICH-coated threads between sister centromeres. The manuscript has to be substantially amended before publication.

We thank the reviewer for this comment. As presented below, we did additional experiments and demonstrated that Topo II α inhibition was leading to an aberrant UFB resolution and, thus, we have substantially modified our first claim. Concerning the details of UFB analysis, we now changed "UFBs" by "PICH-coated UFBs" or "PICH-positive UFBs" in the text, the figures and figure legends and, when we analyzed specific UFB population, we indicated CREST-positive UFBs or FANCD2-positive UFBs (Figures S1A-B and S2B-C).

Metaphase DNA bridges correspond to all short PICH-coated threads. Indeed, to evaluate UFB resolution we quantified UFBs from metaphase to telophase. UFBs are formed when sister chromatid segregation occurs, i.e. during metaphase-anaphase transition (Chan et al., 2007; Baumann et al., 2007; Chan et al., 2009; Chan & Hickson, 2011); all UFBs are thus formed during metaphase when sequences such as centromeres and fragile sites are not decatenated by Topo II α and then become more and more visible through anaphase progression. We thus

assume that UFBs observed in metaphase cells represent the global population of PICH-positive UFBs.

Major concerns:

1. The authors claim that TOP2A inhibition, even throughout the entire mitosis, increases the number of UFBs but does not alter their resolution kinetics, suggesting that TOP2A is dispensable for their resolution. I don't think they can make such a strong claim. The disappearance of UFBs with anaphase-telophase progression could be a result of DNA breakage by mechanical forces or improperly cleaved by other nucleases. Or, though I think is unlikely, PICH may dissociate from UFBs during mitotic exit. However, the authors try to demonstrate that without TOP2A in mitosis, there are no dose-dependent increases in DNA damage (due to chromatin bridges or UFBs) in G1 daughter cells induced by ICRF159 (Figure 4A-C). The main problem of this G1-damage assay is that their counting may be contaminated by other G1 cells that never experience TOP2A inhibition in mitosis. Figure 4A is misleading. Are they started with G1/S synchronised cells? I assume the authors use asynchronous cells treated with EdU for 1 hour, then wait for 8-10hours before the addition of ICRF159 for 1hour. Given that at the beginning there are already cells in mid, mid-late or late S-phase, these EdU-labelled populations will become G1 populations before the ICRF159 treatment. If this was the case, the authors were counting many untreated G1 cells. So the authors have to check and show what is the proportion of EdU+ve G1 cells before the addition of ICRF159? The results need to be subtracted by this population (they did not experience TOP2A inhibition). Besides, the authors did show there is an increase in DNA damage in the next G1 after TOP2A inhibition. This would suggest that TOP2A plays a role during chromosome segregation.

Therefore, although the claim is interesting, I don't think at the moment there is enough evidence to support it.

We thank the reviewer for this important remark. To answer to this question, we synchronized cells by a double thymidine block at G1/S boundary, and the synchronized cells were left untreated or treated with 1 or 10 μ M ICRF-159 during S-phase or during mitosis. Then, we analyzed 53BP1 foci in the next G1. As anticipated by the reviewer, we found that Topo II α

inhibition during S-phase led to a slight, but significant, dose-independent increase in the number of 53BP1 foci (Figure 4 B-C). However, in cells treated with Topo II α inhibitor only during mitosis, we observed a much higher dose-dependent increase in 53BP1 foci. These results demonstrate that DNA damage in G1 was coupled to UFB frequency in the previous mitosis meaning that more UFBs are formed in the previous mitosis, more 53BP1 foci are generated in the next G1. This observation indicates that UFB resolution during anaphase is impaired when Topo II α is inhibited, leading to UFB breakage and DNA damage accumulation in the next G1. These results show that Topo II α activity is necessary for damage-free resolution of UFB during mitosis. All these new results are now presented in Figures 4 and S2A, and the text has been changed accordingly.

2. TOP2A inhibition in mitosis induces UFBs. In Figure 3F, it is not surprising to see overall increases in UFB formation in anaphase. However, it is misleading to compare between metaphase and anaphase because, in metaphase, the authors can only measure centromeric short UFBs whereas I assume the authors counted overall (CEN and arm) UFBs in anaphase. So I suggest the authors clearly state which UFBs they measure as that will change the interpretation significantly.

We understand the reviewer's point and we explained better our rationale in the revised version of the manuscript.

Briefly, in Figure 3F we analyzed average number of UFBs using PICH staining to visualize all types of UFBs. As explained above, UFBs are formed when sister chromatid segregation occurs, i.e. during metaphase-anaphase transition (Chan et al., 2007; Baumann et al., 2007; Chan et al., 2009; Chan & Hickson, 2011), all UFBs are thus formed during metaphase when sequences such as centromeres and fragile sites are not decatenated by Topo II α and then become more and more visible through anaphase progression. We thus assume that UFBs observed in metaphase cells represent the global population of UFBs and, thus, that it is possible to compare the frequency of UFB population in metaphase with UFBs observed latter on in anaphase. Moreover, in our experiments, we observed either an increase (Figure 3F) or a decrease (Figure 3E) in the average number of UFBs per cell between metaphase and early anaphase, indicating that we have no major bias in our analyses.

3. I think the whole study should be just focused on whether S-phase inhibition of TOP2A induced unreplicated DNA bridges in anaphase. The MiDAS data suggest that there is replication stress or underreplication. However, I am not convinced by the assay of unreplicated centromeres. Is this meant in untreated cells, there are always five centromeres unreplicated?

*We understand the reviewer's comment; however, we have several arguments showing that measuring centromeres dots in mitosis is a good readout for centromere replication completion. Indeed, we already extensively described this approach in a previous paper (Gemble et al., PLoS Genet. 2015). We have shown that centromere duplication can be visualized by counting centromere dots using either CREST staining (the approach we used in the present paper) **or by using FISH probes that specifically recognize centromeres**. Both approaches give rise to the same results: we have shown that replication stress leads to an increase in the percentage of single centromere dots, whereas preventing replication stress decreases the percentage of single centromere dots, confirming that counting centromere dots in prometaphase is a good readout for the completion of the replication at the centromeres. Using both approaches (CREST staining and centromeric FISH probes), we observed that a small number of centromeres are not replicated at mitotic entry in unperturbed condition (Gemble et al., PLoS Genet. 2015): these results are likely due to the tumoral status of HeLa cells, used in this paper, known to be genetically unstable. HeLa cells also exhibit a high number of UFBs in unperturbed condition (Figure 1C), confirming that even without any external perturbation, HeLa cells exhibit intrinsic genetic instability, a well-known characteristic of tumor cells.*

Besides, I don't think the authors show MiDAS (EdU foci) at centromeres (Figure 2D & E). I think if the authors can show increased FANCD2 foci formation on mitotic cells and FANCD2-linked UFBs after S-phase inhibition of TOP2A that will further support the claim of arm underreplication. The part of centromere underreplication should be dropped out.

We agree with the reviewer's comment and we quantified the number of FANCD2-associated UFBs, a strong marker of replication stress, in synchronized cells left untreated or treated with

ICRF-159 during S-phase or during mitosis (Figure S2B-C). We found that cells present a significant increase in the percentage of FANCD2-positive UFBs when treated with Topo II α inhibitor during S phase, whereas no increase in the percentage of FANCD2-associated UFBs was observed when the cells were treated during mitosis (Figure S2B-C). Together with MIDAS and centromere replication experiments, these new data clearly confirm that replication stress is generated upon Topo II α inhibition during S-phase.

Therefore, I suggest the authors should focus on the findings of different UFBs formation after cell-cycle specific inhibition of TOP2A or TOP2A inhibition in S phase induces DNA under-replication and UFB formation. The claim that TOP2A is dispensable for UFB resolution should be dropped.

We dropped out the claim that Topo II α is dispensable for UFB resolution in the title, the abstract and the text, as suggested by the reviewer.

Thanks to the reviewer's comment, we performed new experiments with synchronized cells and, as developed in our response to the point 1, we demonstrated in synchronized cells treated with Topo II α inhibitor only during mitosis, a high dose-dependent increase in 53BP1 foci. These results demonstrated that DNA damage in G1 was coupled to UFB frequency in the previous mitosis, indicating a damage-associated resolution of UFB during anaphase when Topo II α was inhibited.

Minor concerns:

1. Can the authors try to explain why there is no dose-dependent increase in UFB formation when TOP2A is inhibited during S phase?

Topo II α levels increase in mid-S phase through mitosis and rapidly decrease upon mitotic completion (Lee and Berger, 2019, Genes ; PMID: 31671531). Thus, 1 μ M of ICRF-159 is probably sufficient to inhibit the whole active Topo II α expressed in S phase, but not in mitosis. This is the reason why there is no dose-dependent increase in UFB formation when Topo II α is inhibited in S phase.

2. page 6, Gemble et al... ref format incorrect.

We agree with the reviewer's comment and we modified the text.

3. page 7, ...concentration, suggesting that

We agree with the reviewer's comment and we modified the text.

4. Figure 2E, y-axis percentage of metaphase cells have >1 EdU foci?

As suggested by the reviewer, in Figure 2E and 3D, we changed the y-axis to "percentage of metaphase cells with at least 1 EdU foci".

5. Figure 3A and 4A legends, please state clearly the treatment time and release time information.

We agree with the reviewer's comment (also raised by the reviewer 2). We thus modified the figure and its legend, and the text to better explain the experimental approach for the treatments presented in Figure 3A and 4A.

6. Figure 3E, are they counting CEN-specific UFBs?

In Figure 3E, we quantified the mean number of all types of UFBs. We now indicated that we quantified the PICH-coated UFBs.

7. ICRF drugs can cause DNA damage by inhibiting TOP2B, so the authors should elaborate the possible reasons why ICRF159 treatment in S phase may induce incomplete DNA replication.

Several studies in vitro or in yeast already reported that Topo II α is necessary for DNA replication during S-phase (Baxter, 2015; Bar-Ziv et al., 2016; Bailey et al., 2015; Ishimi et al., 1992; Gaggioli et al., 2013). Our data suggest, thus, that Topo II α could have a similar role in mammalian cells explaining why Topo II α inhibition in S-phase impairs completion of DNA replication. This was already mentioned in the first version of the manuscript ("These data

confirm that Topo II α inhibition results in an accumulation of unreplicated DNA during mitosis, reflecting incomplete DNA replication in the previous S-phase. These observations are consistent with several studies in yeast or in vitro, showing that Topo II α facilitates DNA replication [2, 27-30]. They also suggest that Topo II α activity is essential to promote complete DNA replication in mammalian cells”, page 7). More importantly, Topo II α is 18 times more sensitive to ICRF-159 compare to Topo II β (Perrin et al., 1998, Biochem Pharmacol; PMID: 9763227). In consequence, we assume that the effect of the ICRF-159 treatment on DNA replication is mainly due to the inhibition of Topo II α activity rather than Topo II β activity.

Referee 2:

Comments to the Author(s)

It has been conventionally thought that since inhibition of Topoisomerase II α (Topo II α) leads to a large increase in Ultra Fine Bridges (UFBs), its activity was required for UFB resolution. However this manuscript by Amor-Gueret's group have put forward the hypothesis that the increase in UFB frequency upon TopoIIa inhibition, could be a result of just the accumulation of newly formed UFBs. The authors argue that lack of TopoIIa activity prevents UFB formation in a cell cycle specific manner and is not required for UFB resolution. The authors have argued out their case with a certain amount of conviction. The experiments are reasonably well done and controlled. The manuscript however requires the following modifications, additions before it becomes acceptable.

1. For all the figures the authors have used experiments based on cell cycle. Histograms should be provided for each experimental condition and treatment to show the cell cycle stage in which the experiments were conducted.

Initially, we performed all experiments using asynchronous cells and based on the known duration of each phase of HeLa cell cycle (Hahn et al., 2009). Indeed, it is known that HeLa cells take about 8-10 hours between S-phase to mitosis (Hahn et al., 2009), meaning that by analyzing cells in mitosis that were treated during 8 hours, we analyzed cells that were at the beginning of S-phase when we added ICRF-159. We modified the text and the figure 1A to clarify this point.

Moreover, as suggested by the reviewer 1, we reproduced the experiments presented in Figure 3A-C using cells synchronized in G1/S by double thymidine block and then released and treated with ICRF-159 during either S-phase or mitosis (cell cycle distribution of HeLa cells synchronized in G1/S using double block thymidine and then released and left untreated or treated with 1 or 10 μ M ICRF-159 during S-phase or during mitosis is shown in Figure S2A). These results, presented in Figure S2A-C of the new version of the manuscript, confirmed that in synchronized cells UFB frequency is increased in both cells treated with ICRF-159 during S phase or during mitosis.

2. For the main datasets (i.e. the proof of principle experiments) – siRNA to TopoII α should be conducted so that the data obtained with ICRF-159 can be validated.

The use of siRNA to deplete Topo II α could have been an attractive approach to confirm our observation. However, because this is not possible to temporally control when Topo II α will be depleted by siRNA, we would be not able to specifically deplete Topo II α either during S-phase or during mitosis. For this reason, we performed all our experiments using specific Topo II α inhibitor widely used in the literature, allowing us to precisely and temporally control when we inhibited Topo II α activity.

3. The authors in Page 5 state: by treating the cells with ICRF-159 from S-phase until the end of mitosis”. More clarity is required – which stage of S-phase? Early, middle or late? Exact number of hours of treatment should be indicated in the figure and in the histograms mentioned in point #1.

We understand the reviewer’s comment and we modified the text and the figure 1A to better explain our experimental approach. HeLa cells have a very well described cell cycle (Hahn et al., 2009). In particular, it has been shown that G1 phase takes about 6 hours, S-phase takes about 6 hours and G2 phase takes about 3 hours. Based on these results and because we analyzed cells in mitosis, we assumed that cells treated during 8 hours and analyzed in mitosis are cells that were at the beginning of S-phase when treated with ICRF-159. Moreover, as previously mentioned, we also reproduced our experiments using cells synchronized in G1/S by double thymidine block and then released and treated with ICRF-159 during either S-phase or mitosis (new Figure S2A). This complementary approach clearly confirmed that Topo II α inhibition leads to UFB formation in a cell cycle dependent manner (new Figure S2B).

4. The authors have used two different concentrations of ICRF-159. Yet, except for UFB formation, two different concentrations of ICRF-159 have the same effect. The authors should either give a plausible reason or do the experiments using a gradient of ICRF-159 where differential effect will be visible.

Topo II α levels increase in mid-S phase through mitosis and rapidly decrease upon mitotic completion (Lee and Berger, 2019, Genes ; PMID: 31671531). Thus, 1 μ M of ICRF-159 is probably sufficient to inhibit the whole active Topo II α expressed in S phase, but not in mitosis. This is the reason why there is no dose-dependent increase in UFB formation when Topo II α is inhibited in S phase.

5. For the amount of data presented in the manuscript, four Figures are not necessary. The authors should try to put their data in Figure 4 as part of Figure 3.

We understand the reviewer's point. However, the results presented in figures 3 and 4 are related to two distinct messages. Indeed, in figure 3, our results show that Topo II α inhibition leads to the formation of two types of UFBs depending on the cell cycle phase. In figure 4, our data decipher the role of Topo II α inhibition on UFB resolution and its consequence on DNA damage inheritance in the next G1. For this reason, we think that keeping figures 3 and 4 separated is better for the clarity of the paper. Moreover, in the new version of the manuscript we added some additional figures (S1 and S2) making difficult the merge of figures 3 and 4.

6. The authors should try to tighten the language in the text for Figure 3. It is extremely rambling and verbose for the moment.

We agree with the reviewer's comment and we modified the text.

Appendix B

Response for the revision of the article:

Manuscript ID: RSOB-19-0259

Title: TOPOISOMERASE II α PREVENTS ULTRAFINE ANAPHASE BRIDGES BY TWO
MECHANISMS

Authors: Gemble et al.,

Corresponding authors: mounira.amor@curie.fr and simon.gemble@curie.fr

Dear Editor,

We would like to thank you and to thank Open Biology editorial board for accepting our manuscript.

Below you will find our detailed response for the revision of the article. Our comments are in blue italic.

In this final revised version, we modified the text to better improve the understanding of our study and to specifically address the additional comments from reviewer 2.

Moreover, we addressed all the editorial requests to ensure that our paper complies with all of guidelines.

We do hope this improved version of the manuscript is suitable for publication in Open Biology.

We are looking forward to hear from you soon,

With kind regards,

Mounira Amor-Gu ret and Simon Gemble

The authors have carried out a new and important experiment to re-test if TOP2A is dispensable for proper UFB resolution during mitosis. In contrast to their previous conclusion, they now demonstrate that TOP2A is required for proper or damage-free UFB resolution, as published in other previous studies. Regarding to another claim that TOP2A didn't affect the kinetics of UFB resolution, I found that their data are not consistently supporting it. In fig.1D, ICRF caused increases of UFB formation in early anaphase (~21/cell), which reduced to ~13/cell in late anaphase, indicating that cells cannot timely resolve all UFBs without TOP2A. However, in figs 3E and F, both showed a very sharp reduction of UFBs in late anaphase, arguing that TOP2A is dispensable. In fig. 3F, even cells have a higher number of UFBs in early anaphase (~29/cell), the drop is even more obvious (to ~6/cell in late anaphase). According to the result of fig. 1D, it suggests that there is a defect in UFB resolution kinetics whereas not from figs 3E and F.

We understand the reviewer's comment, but we think that it is difficult to strictly compare the values obtained in the figures mentioned by the reviewer. Indeed, data presented in figures 1D, 3E and 3F do not correspond to the same experimental approach. In Figure 1D, cells were treated with ICRF-159 during both S-phase and mitosis. In figure 3E and F, cells were treated only during S-phase or during mitosis, respectively. However, we consider that the global disappearance of UFBs can be compared. Indeed, we showed that even if Topo II α is inhibited during mitosis (Figure 1D and 3F), the global disappearance of UFBs is observed during mitotic progression in both treated and untreated condition.

To avoid any misunderstanding we modified the text, changing "kinetics of UFB resolution" by "global UFB disappearance".

On one hand, the authors now show TOP2A confers proper UFB resolution during mitosis but on the other hand, they claim the lack of TOP2A does not affect the kinetics of UFB resolution/disappearance (data seem not consistently support it). It is very hard to rationalize their conclusion. The authors should give a more reasonable explanation, or their interpretation may be incorrect.

As already explained in the manuscript and in the comment above, we demonstrate that Topo II α is necessary for proper UFB resolution but not for the global disappearance of UFBs during mitosis. On the basis of our results showing a dose-dependent accumulation of 53BP1 foci in the next G1 phase of synchronized cells treated with ICRF-159 during mitosis, we propose that UFBs are resolved in the absence of Topo II α activity through an aberrant pathway associated with the transmission of DNA damage in the next G1, likely by DNA breakage.

Besides, the authors assume that the counting of metaphase UFBs can be used to represent the global UFB population in anaphase. It is totally misleading. As mentioned before, in metaphase where sister chromatids cohesion remains present, UFB can only be visualised at centromeres, so it is impossible to see catenane UFBs on arms. In anaphase where sister chromatid arms are separated, the authors can count all UFBs both from arms and centromeres. Therefore, it is inappropriate to compare UFB formation and resolution kinetics between metaphase and anaphase cells, unless they only count centromeric UFBs in anaphase; however, they didn't do it in Fig. 1D, 3E, 3F. So, the kinetics can only be determined from early anaphase to telophase. Please provide evidence and image examples showing where UFBs form in metaphase cells, ideally with a CREST staining.

To our knowledge, it has not been reported that UFBs on chromosome arms could not be detected in metaphase. However, since we now compare the “global UFB disappearance”, instead of the “kinetics of UFB resolution”, we think that this point is minor.

In page 9 (Fig. S1B), it is inaccurate to claim that most of UFBs induced by ICRF are of centromeric origin. The treatment of ICRF induces overall numbers of UFBs (both on arms and centromeres). The UFB CEN to ARM ratio only slightly increased and is not in a dose-dependent manner. Overall, I do not totally agree with the interpretation of some data and I think the authors should revise their claims carefully before accepted for publication.

We agree with the reviewer comment and we modified the text in the new version of the manuscript: “Consistent with this hypothesis, we observed that 66% of UFBs were of centromeric origin when Topo II α was inhibited during mitosis”. We agree that the percentage of CREST-positive UFBs was the same when cells were treated with 1 μ M or 10 μ M ICRF-159. However, 66% of CREST-positive UFBs correspond to 5.016 UFB per anaphase cell in response to 1 μ M ICRF-159 treatment during mitosis, whereas they correspond to 12.54 UFB per anaphase cell in response to 10 μ M ICRF-159 treatment during mitosis, which is more than twice higher.